# Genetic Improvement of Wheat with Pre-Harvest Sprouting Resistance in China

**DOI:** 10.3390/genes14040837

**Published:** 2023-03-30

**Authors:** Cheng Chang, Haiping Zhang, Jie Lu, Hongqi Si, Chuanxi Ma

**Affiliations:** Key Laboratory of Wheat Biology and Genetic Improvement on Southern Yellow and Huai River Valley, Ministry of Agriculture and Rural Affairs, College of Agronomy, Anhui Agricultural University, Hefei 230036, China

**Keywords:** common wheat, pre-harvest sprouting, gene, germplasm resources, wheat variety breeding

## Abstract

Wheat pre-harvest sprouting (PHS) refers to the germination of seeds directly on the spike due to rainy weather before harvest, which often results in yield reduction, quality deterioration, and seed value loss. In this study, we reviewed the research progress in the quantitative trait loci (QTL) detection and gene excavation related to PHS resistance in wheat. Simultaneously, the identification and creation of germplasm resources and the breeding of wheat with PHS resistance were expounded in this study. Furthermore, we also discussed the prospect of molecular breeding during genetic improvement of PHS-resistant wheat.

## 1. Introduction

Wheat is one of the major food crops worldwide, accounting for 20% of humans’ total calories and protein intake, and feeds nearly 40% of the global population. A stable increase in wheat yield per unit and total yield is essential to cope with the increasing demand of feeding the population and is important for ensuring wheat production safety [1,2]. Pre-harvest sprouting (PHS) in wheat is a climatic disaster, which refers to seeds directly sprouting on the spike because of rainfall before harvest. PHS often occurs in China, the United States of America, the European Union, Canada, Australia, Japan, and other countries, resulting in a minimum loss of 1 billion U.S. dollar globally every year [3,4]. Because of the hydrolysis of endosperm storage materials (such as starch and protein), the 1000-grain weight and bulk weight of germinated seeds decrease, resulting in yield reduction. Furthermore, when the activity of *α*-amylase in germinated grains increases, the end-use quality of wheat deteriorates because of starch and protein decreases [3,5,6,7]. At the same time, the seedling quality is negatively affected because of the decreased seed vigor [5,7]. Therefore, PHS is one of the climatic disasters affecting the safety of wheat production worldwide and has been studied by the major wheat-producing countries [7,8].

PHS in wheat is a complex trait that is affected by internal factors such as seed dormancy level, seed coat color, ear and glume characteristics, and seed development stage, as well as external factors such as weather [9,10,11]. It is generally believed that red-grained wheat has higher PHS resistance than white-grained wheat; however, the latter has better grinding qualities (such as high flour yield and favorable color); therefore, it is often favored by producers and consumers [7,12]. Because the resistance of modern white-grained varieties toward PHS is generally weak, they are more seriously affected by PHS than red-grained varieties [5]. Seed dormancy is the main genetic component for PHS resistance and determines the resistance level of wheat. Therefore, the genetic regulation mechanism of seed dormancy is often taken as the main content when studying the mechanism of PHS resistance in previous reports. Many studies have found that PHS resistance/seed dormancy is controlled by multiple genes, which are distributed across 21 chromosomes of wheat [13,14,15,16,17]. Therefore, the leading approaches and strategies to combat the risk of PHS involve breeding and applying the PHS-resistant wheat variety. Therefore, excavating and identifying the important genes/loci, transferring, and pyramiding or editing major genes will be the main area of research in molecular breeding for improving PHS resistance.

## 2. The Quantitative Trait Loci for PHS Resistance

PHS resistance in wheat is a complex trait controlled by multiple genes, where seed dormancy is the main genetic component [13,14,15,16,17]. Previous studies have mainly used two methods to excavate and identify quantitative trait loci (QTLs) for this trait. One method is linkage analysis based on biparental populations such as recombinant inbred line (RIL), double haploid (DH), backcross, backcross inbred line (BIL), near-isogenic line (NIL), and F2, whereas the other method is whole-genome association analysis based on a panel of varieties. Therefore, many QTLs for PHS resistance have been identified through different methods, which are widely distributed across 21 chromosomes [13,14,15,16,17,18,19]. With the development and application of molecular marker identification techniques, such as SNP chip (90 k, 660 k, etc.), more and more gene loci of PHS resistance/seed dormancy have been identified, providing more effective markers for marker-assisted selection during genetic improvement of PHS resistance [13,14,15,16,17,18,19,20,21,22,23,24,25,26,27,28,29,30,31]. Owing to the limitations of the methods, many identified QTLs are material specific and not reproducible in different studies; therefore, they have certain limitations in marker-assisted selection during wheat breeding for PHS resistance. Based on both the previous studies and our own research, we found that the QTLs located on chromosomes 3 and 4 played a significant role in PHS resistance. This was especially the case for the QTLs on chromosomes 3A and 4A (Table 1), which showed good stability in multiple environments. These loci of PHS resistance are important for gene cloning and marker-assisted breeding. 

## 3. The Regulation Genes for PHS Resistance

The exploration and identification of genes is the basis of studying the molecular regulation mechanism of PHS resistance in wheat. To date, only a few functional genes have been identified for PHS resistance. So far, seven genes have been isolated through homologous cloning and mapping cloning: *TaVp-1* [38,39,40,41,42], *TaDOG1L1* [43], *TaMFT*/*TaPHS1* [8,44], *TaSdr* [45], *TaMKK3* [46], *TaQsd1*/*TaAlaAT* [47], and *Myb10-D* [6] (Table 2). These genes mainly participate in the regulation of seed dormancy by affecting the ABA/GA synthesis/metabolism/signaling pathways, thereby affecting PHS resistance. This indicates that ABA/GA is a key factor involved in the regulation of seed dormancy/PHS resistance. The application of these genes is different in molecular breeding for PHS resistance. At present, *TaVp-1*, *TaMFT*/*TaPHS1* and *Myb10-D* are widely used in wheat PHS resistance breeding. However, with the deepening of molecular genetics research on PHS resistance, more and more candidate genes have been excavated and verified, and these will provide more gene resources for wheat PHS resistance breeding.

In our study, along with the methods of homologous cloning and mapping cloning, some candidate genes involved in the regulation of PHS resistance in wheat were also excavated using bioinformatics and transcriptomic tools. These genes include *TaGASR34* gene (Gibberellic Acid-Stimulated Regulator gene family), *TaC3H4*/*18*/*37*/*51*/*72* genes (C3H zinc finger transcription factors gene family), *TaGATA17*/*25*/*34*/*37*/*40*/*46*/*51*/*72*/*73* genes (GATA transcription factors gene family), *TaVQ* genes (valine–glutamine proteins), *TaIQD*/*4*/*28*/*32*/*58*/*64*/*69*/*71* genes (IQ67 Domain gene family), *TaCDPK21* (calcium signaling pathway gene) and *TaCPK40* (calcium-dependent protein kinases gene) [50,51,52,53,54,55,56]. The roles and functions of these candidate genes in PHS resistance and their breeding value should be studied in the future.

## 4. Germplasm Resources of PHS-Resistant Wheat in China

Pre-harvest sprouting of wheat often occurs in the southwest winter wheat region (including Sichuan, Yunnan, Hunan, Hubei, and Guizhou, etc.), the middle and lower Yangtze River winter wheat region, and the north and northeast spring wheat regions in China. Due to long-term mechanisms of natural and artificial selection, there are many germplasm resources with PHS resistance in these areas. Some of the varieties with high PHS resistance in the southwest winter wheat region include Wanxianbaimaizi, Yongchuanbaimaizi, Suiningtuotuomai, Fulingxuxumai, Baikezao 2, Chuanmai 11, Mianyang 115, Chuanyu 2179, Guimai 2, and Mian 84-43. The PHS-resistant wheat germplasms in the middle and lower Yangtze River winter wheat region include Dahuangpi, Jiangsubaihuomai, Ningmai 8, Yangmai 10, Yangmai 158, Yangmai 20, and E ’en 5. In the north and northeast spring wheat region, Kefeng, Kehui, Kejin 1, Liaochun 2, Kehan 6, Daqingmang, Jinghong 8, Neimong 5, Longmai 26, and Longfumai 5 have high PHS resistance. PHS resistance is especially prominent among landraces and local wheat varieties [7].

Due to climate change, the Huang–Huai winter wheat area, which is the main wheat-producing area of China, is prone to frequent precipitation during the harvest period. PHS often occurs because semi-winter white-grained wheat varieties with weak PHS resistance are grown in this area, which significantly affects the production safety of the wheat. In this area, some varieties have acquired favorable PHS resistance, including Lanhuamai, Huixianhong, Baihuomai, Baiyupi, Neixiang 19, Neixiangbodijiang, Xiaoyan 5, Yumai 18, Bainong 64, and Bainong 3217 [7,57]; however, most of them are landraces, local varieties, or old varieties. Modern wheat varieties with high PHS resistance are relatively few in number. PHS-resistant germplasm resources in this region included mainly landraces or local varieties, which were eliminated because of their poor yield and agronomic traits. Because of the poor agronomic and yield characters of most local varieties or landraces, these wheat varieties are seldom utilized in production; instead, they are mostly used as parents in PHS resistance breeding. In the Huang–Huai wheat area, there are few white-grained varieties with favorable PHS resistance. Most are generally weak in PHS resistance. Because of the rainfall before harvest, the harm caused by PHS is more pronounced, making it one of the main negative factors affecting the safety of production in this wheat area. However, germplasm resources or modern varieties, such as Yumai 18, Bainong 64, Yumai7698, and Bainong 3217, are more suitable for the present requirement of PHS resistance breeding because of their favorable yield and agronomic traits. The Huang–Huai wheat area is one of the main wheat areas in China, and the few varieties with favorable PHS resistance cannot widely meet the needs of wheat safety production in this area. Therefore, the main way to solve the harm caused by PHS in this area is to strengthen the breeding of PHS resistance and cultivate more white-grain varieties with good PHS resistance.

In this study, a few advanced wheat lines with good PHS resistance were selected from crosses of the medium PHS resistance varieties Annong 0711(moderately resistant to PHS, GI: 21.6%) and Weilai 0818 (moderately resistant to PHS, GI: 27.2%) using phenotypic and molecular marker-assisted selections, according to the method in our previous study [58]. These lines are semi-winter white-grained wheat and will have good application prospects for PHS resistance breeding in the Huang–Huai wheat region (Table 3). These results also indicate that wheat PHS resistance can be significantly improved through cross breeding and marker-assisted selection, as well as providing a technical reference for wheat PHS resistance breeding.

## 5. Progress in Breeding PHS-Resistant Wheat in China

The winter wheat areas, including Chinese southwest, the middle and lower reaches of the Yangtze River, as well as the spring wheat areas, including the northeast and the north, often experience rainy weather before wheat harvest, which facilitates PHS occurrence. In these wheat areas, red-grained spring wheat varieties are often developed and planted to resist PHS. Therefore, breeding wheat with high PHS resistance has always been one of the main goals in these regions. The breeding methods include traditional breeding, such as crossing and backcrossing between wheat parents (a method majorly used in China), and molecular breeding for PHS resistance (less used).

### 5.1. Traditional Breeding for PHS-Resistant Wheat

Under strict selection pressure (long-term natural and artificial selection), many wheat varieties with good PHS resistance have been identified and bred by the conventional hybridization method in PHS-prone areas. The resistance to PHS of these varieties is generally better than that of the varieties grown in the Huang–Huai and northern winter wheat regions. In the southwest winter wheat region, PHS-resistant varieties include Wanxianbaimaizi, Yongchuanbaimaizi, Fulingxuxumai, Chuanmai 11, Yonggang 2, Guimai 2, Mian 79-2, Chuanyu 26, Chuannong 16, Nanmai 991, and Mianmai 43 [7,59,60]. The varieties with high PHS resistance in the middle and lower Yangtze River winter wheat region include Dahuangpi, Jinda 2905, Ningmai 8, Ningmai 26, Yangmai 10, Yangmai 20, Yangmai 24, and Zhenmai 10 [7,61]. The varieties with high PHS resistance in the spring wheat areas of north and northeast China include Songhuajiang 1, Kefeng, Liaochun 1, Fengqiang 5, Neimai 17, Longfumai 5, Kechun 19, Longmai 39, and Longmai 72 [7,62,63,64], as shown in Appendix A. Most of these varieties are red-grained spring varieties, including landraces, local varieties, and modern varieties developed more recently. However, most of the varieties with favorable PHS resistance are red-grained, while few white grained wheat varieties are cultivated and applied in these areas. The resistance mechanism is closely related to grain color: in other words, compared with the white-grained varieties, the red-grained varieties have more advantages in terms of PHS resistance, and this advantage has been maintained until now.

In the Huang–Huai and northern winter wheat areas, mainly white-grained semi-winter and winter wheat varieties are planted, and the PHS resistance of these varieties, especially of modern varieties, is generally poor. With climate change, the harm caused by PHS in these two wheat areas is becoming increasingly serious; therefore, many wheat breeders have gradually shifted their attention to improving PHS resistance. Some landraces and local wheat varieties with good PHS resistance are Sanyuehuang, Lanhuamai, Zaoyangmai, Bima 1, Hulutou, Beijing 15, and Baiyupi [7]. Generally, there were also some PHS-resistant modern varieties, including Yumai 18, Bainong 64, Shanxi 229, Lumai 21, Bainong 3217, Xiaoyan 6, Weilai 0818, Zhengmai 7698, Lumai 47, Annong0451, and Annong0805 [7,41,57]. The PHS resistance of these varieties reached a medium level, which was generally lower than that of the varieties in the southwest, middle, and lower reaches of the Yangtze River winter wheat region, as well as north and northeast spring wheat regions. However, the application of these varieties plays an important role in alleviating the harm caused by PHS in these wheat regions. As a result of climate change, the Huang–Huai winter wheat area is becoming increasingly prone to rainfall before harvest. Due to the lack of white-grained wheat varieties with high PHS resistance, the damage caused by PHS in this region is becoming more and more serious, and sometimes the loss is greater than that in traditionally PHS-prone wheat areas, such as the southwest winter wheat region (including Sichuan, Yunnan, Hunan, Hubei, and Guizhou, etc,), the middle and lower Yangtze River winter wheat region (including Jiangsu, Anhui, Hubei, etc.), and north and the northeast spring wheat regions (including Heilongjiang, Liaoning, Jilin, and Inner Mongolia) in China. It is necessary to breed and apply wheat varieties with strong resistance to PHS.

### 5.2. Molecular Marker-Assisted Selection Breeding for PHS-Resistant Varieties

Many previous studies have shown that the QTLs for PHS resistance were almost distributed across 21 chromosomes of wheat; seven regulatory genes were identified and several molecular markers were developed to use during PHS resistance breeding. However, most PHS-resistant varieties were bred by the conventional crossing of parents, and only a few varieties were developed by molecular breeding approaches. Additionally, a few genes/loci, mainly distributed on chromosomes 3A, 3B, and 4A, were effectively used in PHS resistance breeding (Table 1 and Table 2). Through the cross between Zhongmai 18 (PHS-resistant, GI: 20.1%) and Jining 13 (moderately PHS resistant, GI: 32.3%), Xiao et al. (2014) developed a white-grained semi-winter wheat variety with high PHS resistance, Zhongmai 911, using molecular marker-assisted selection of Barc 310 on 3AS [65]. The variety contained the favorable allele (*Barc321-199* bp) for PHS resistance and had a lower GI (13.2%) compared with its parents (Table 4).

In our previous study [58], we pyramided the genes/loci on 3AS (*Barc321a*/*TaMFT-3Aa*), 3BL (*TaVp-1Be*), and 4AL (*TaMKK3-Ab*) from the parents Aizao 781 (moderately resistant to PHS, GI: 32.4%), Bainong 64 (moderately resistant to PHS, GI: 37.2%), and Yannong 19 (sensitive to PHS, GI: 71.6%). Through marker-assisted selection and phenotypic identification, the wheat variety Annong 0711 (moderately resistant to PHS, GI: 21.6%), which carried the favorable alleles on 3AS, 3BL, and 4AL [58], was developed with a higher PHS resistance than that of its parent varieties. These results indicated that these major genes/loci on chromosomes 3AS, 3BL, and 4AL can be used for breeding PHS-resistant varieties (Table 4). 

From the perspective of the application of PHS-resistant gene/loci in wheat breeding, the gene resources that can be effectively applied are still limited to 3AS (*Barc321*/*TaMFT-3A*), 3BL (*TaVp-1B*) and 4AL (*TaMKK3-A*), while other reported genes/loci are rarely used. The reason may be that the functions of those genes need to be further verified, or they have little effect on PHS resistance and lack effectiveness. However, as more and more candidate genes/loci are excavated and identified, more and more genes will be used in wheat PHS resistance breeding [65].

### 5.3. Transgenic and Gene Editing Breeding for PHS-Resistant Varieties

So far, seven genes for PHS resistance have been identified, which can be used in transgenic and gene editing breeding. Huang et al. (2012) transferred the maize gene *Vp-1*, including promoter and coding regions, into the wheat variety Zhengmai 9023, and the seed GI value of T3–T5 of the transgenic generations decreased by 79%, 80%, and 82% compared with the wild type, respectively [66]. These results indicated that the maize gene *Vp-1* can effectively improve the dormancy of grain and increase PHS resistance in wheat. Clustered regularly interspaced palindromic repeats (CRISPR), a gene editing technology, has made rapid developments in the past few years and has become an important tool for plant functional gene research and crop genetic improvement. The *Qsd1* gene encodes alanine aminotransferase and is an important gene controlling dormancy in barley grains. In our study, *TaQsd1* was obtained by homologous cloning in wheat, and its involvement in the regulation of seed dormancy was confirmed [47]. In a previous study, CRISPR-Cas9 technology was used to edit *TaQsd1*, and the genetically edited wheat had a significantly longer dormancy period and improved PHS resistance compared with the wild type [67]. In China, using CRISPR-Cas9 technology, Zhu et al. (2022) restored *Tamyb10* in the white-grained variety Fielder to create PHS-resistant wheat. In their research, the transgenic plants with edited *Tamyb10-B1a* showed a higher elevated anthocyanin index of seed and PHS resistance than the wild type (Fielder), and the grain color changed from white in the wild type to red in the edited type [68]. Previous studies have shown that the application of gene-editing technology can effectively improve breeding progress of PHS resistance in wheat. Therefore, gene editing may be an efficient breeding technique for improving wheat PHS resistance. We also identified some genes and their dominant alleles for PHS resistance, including *TaVp-1A* (*TaVp-1Ab*, *TaVp-1Ad*), *TaVp-1B* (*TaVp-1Bb*, *TaVp-1Bc*, *TaVp-1Bd*, *TaVp-1Be*, and *TaVp-1Bf*), *TaMFT-3A* (*TaMFT-3Aa*), and *TaQsd1* (*TaQsd1-5Ba*) [39,40,41,47,48]. Simultaneously, some candidate genes, such as *TaGASR34*, *TaC3Hs*, *TaGATAs*, *TaVQs*, *TaIQDs* and *TaGA20ox1*, were identified successively in our study, and can be considered as important gene resources for transgenic breeding or gene editing/breeding for improving wheat PHS resistance after further verifying their functions [50,51,52,53,54,55,56].

## 6. Conclusions and Perspectives

The main way to reduce the harmful effects of PHS is to breed new wheat varieties with favorable PHS resistance. Future studies should focus on detecting and identifying major genes/loci, creating and applying PHS-resistant germplasm resources, and improving breeding technology. Although many genes/loci associated with PHS resistance have been identified across 21 chromosomes, only a few have been applied in wheat breeding in China. The genes/loci on chromosomes 3A, 3B, and 4A are used more often than the others, which suggests that the other genes/loci may have low effects on PHS resistance in wheat. Therefore, it is particularly necessary to use forward and reverse genetics, bioinformatics, transcriptome analyses, and other approaches to detect major genes/loci for PHS resistance. Regarding germplasm resources, PHS-resistant, red-grained semi-spring varieties are relatively rich germplasm resources, which are mainly distributed in the winter wheat region of southwest, the middle and lower reaches of the Yangtze River, and the spring wheat region of north and northeast of China. However, white-grained semi-winter wheat varieties are mainly planted in the Huang–Huai wheat region. The lack of favorable PHS-resistant germplasm resources harms the breeding of PHS-resistant varieties in this wheat region. Simultaneously, this lack of white-grained semi-winter varieties with high PHS resistance cannot fulfill the production requirements of the Huang–Huai wheat region. Therefore, identification of the genes/loci, as well as screening and production of germplasm resources, are crucial for genetically improving PHS-resistant wheat varieties.

To date, most breeding methods for PHS resistance involve traditional hybridization between the parents instead of molecular breeding. Previous studies have shown that molecular marker-assisted selection can effectively improve the efficiency and accuracy of breeding PHS-resistant wheat varieties such as Zhongmai 911 and Annong 0711. Other studies have also shown that transplanting genes and gene editing can significantly improve seed dormancy and PHS resistance. The identification of more genes for PHS resistance can broaden the scope of molecular breeding techniques (such as transgenic and gene editing techniques) in breeding wheat varieties. Therefore, it is particularly important to develop and identify excellent germplasm resources with high PHS resistance and to clone some functional genes with significant effects on PHS resistance during molecular breeding of wheat. Presently, transgenic and gene editing-based wheat varieties cannot be released and applied in the market in China. However, as effective molecular breeding techniques, the transgenic and gene editing approaches have received increasing attention for the breeding of PHS-resistant wheat varieties.

## Figures and Tables

**Table 1 genes-14-00837-t001:** The quantitative trait loci for pre-harvest sprouting resistance on chromosomes 3A and 4A.

Chromosome	Molecular Markers	Physical Site/Mb	PVE%	References
3AS	Barc310/Bcd907	7.1	11.6–44.8	[13]
	Barc12/Barc321	11.7.7–15.5	10.1–41.0	[18]
	Barc12/Barc57/ Barc57/Barc321	10.3.2–15.5	-	[15]
	Barc57/Barc294	7.9–10.3	25.6–48.3	[20]
	Barc321/Barc57	10.3.2–11.7	5.7–15	[21]
	Barc310/wPt-4545	7.1	6.5–6.9/ 5.7–6.6	[22]
	wsnp_Ra_c2339_4506620/Xbarc57	10.2–10.3	8.5–8.7/ 13.7–15.9	[19]
	KASP-222	7.3	21.6–41	[23]
	Tdurum_contig83209_316/Kukri_c1568_942	4.2	5–5.4	[24]
3AL	fbb293	-	5.5–7.9/4.5–7.1	[25]
	Bcd1380/fbb370 fbb370/Bcd907	636.7	23.3– 38.2	[26]
	Bcd141/Cdo345	657.9	13	[26]
	Wmc153/Gwm155	701.7–703.0	24.7–35.2	[27]
	Wmc153/Gwm155	701.7–703.0	16	[28]
	Tamyb10-A1-66	704.0	6.8–12.1/ 7.5–11.1	[19]
	AX-109376167/AX-111037462	596.0–596.3	19.3–35.7	[29]
	IWB63289/IWB72045	689.5–741.2	3.4–8.6	[30]
	AX-111578083	704.5	11.5–25.1	[29]
4AL	Cdo795/Psr115	545.6–634.7	33–77.2	[31]
	Cdo795/Bcd808	545.6	12.9–13.6	[13]
	Barc170/Gwm397	607.9	13.1–23.3	[32]
	Gwm397,Barc170,Hbe03	607.9	6.9–43.3	[33]
	Gwm397, Gwm269,Barc170	607.9	25–38	[34]
	Gwm637/Gwm937	-	9.3–40.8/14.3–39.8	[14]
	ZXQ118,Barc170,Wg622,Gwm269	607.9	-	[35]
	Gwm494	634.4	6.3–12	[36]
	Barc170-Gwm397	607.9	2.5–17.8/7–17.5	[37]
	Gwm397	607.9	6.7–10.0	[22]
	Ex_c66324_1151/wsnp_Ex_rep_c66324_64493429	604.2–604.2	9.8–47.6	[19]
	AX-111634210	718.9	19.5–36.4	[29]
	IWB23723	617.0	13.9	[30]

Note: - means no data. PVE: phenotypic variance explained.

**Table 2 genes-14-00837-t002:** The genes and their favorable alleles for pre-harvest sprouting PHS resistance in wheat.

Genes	Chromosomes	Markers	Physical Site/Mb	Favorable Alleles	References
*TaVp-1*	3AL	Vp-1A	659.6	*TaVp-1Ab*, *TaVp-1Ad*	[41]
		Vp1A3		*TaVp-1Agm*	[42]
	3BL	Vp1-b2	693.3	*TaVp-1Bb*, *TaVp-1Bc*, *TaVp-1Bd*, *TaVp-1Be and TaVp-1Bf*	[39,40]
		Vp1B3		*TaVp-1Bb*, *TaVp-1Bc*	[38]
*TaDOG1L1*		DOG1L1	534.5	-	[43]
*TaMFT*/*TaPHS1*	3AS	MFT-A1	7.3	*TaMFT-A1b*	[8]
		MFT-A2		*TaMFT-3Aa*	[48]
		MFT-3A		SNP-222(C)	[44]
		TaPHS1-SNP1/TaPHS1-SNP1		SNP646/666 (G/A)	[5]
*TaSdr*	2A	Sdr2A	158.5	*TaSdr*-*A1a*	[49]
	2B	Sdr2B	200.6	*TaSdr*-*B1a*	[45]
*TaMKK3*	4AL	MKKAC	710.2	*TaMKK3-A1b*	[46]
*TaQsd1*	5B	QSD1	387.7	*TaQsd1-5Ba*	[47]
*TaMyb10-D*	3D	PHS-3D	570.8	*Myb10-D*	[6]

Note: - means no data. PVE: phenotypic variance explained.

**Table 3 genes-14-00837-t003:** The advanced lines of wheat with favorable pre-harvest sprouting resistance.

No.	Names of Lines	Cross of Two Parents	GI Value/%	PHS Resistance Level
1	WA-2	Weilai 0818/Annong 0711	15.1	R
2	WA-19	Weilai 0818/Annong 0711	20.0	R
3	WA-23	Weilai 0818/Annong 0711	10.2	R
4	WA-41	Weilai 0818/Annong 0711	19.3	R
5	WA-115	Weilai 0818/Annong 0711	20.0	R
6	Annong98	Bainng207/Wannong09174	29.4	MR

Note: GI means germination index. R and MR mean resistance and moderate resistance, respectively.

**Table 4 genes-14-00837-t004:** The varieties developed by marker-assisted selection.

Varieties	Parents	Type	Genes/Loci for PHS Resistance	Favorable Alleles	GI Value/%
Zhongmai 911	Zhongmai 18	White-grained semi winter	3AS (*Barc310*)	*Barc310-199bp*	20.1
	Jining 13	White-grained semi winter	-	-	32.2
Annong0711	Aizao 781	White-grained semi sbring	3AS (*Barc321*)	*Barc321a*	32.4
	Bainong 64	White-grained semi winter	3AS/3BL/4AL	*TaMFT-3Aa*/*TaVp-1Be*/*TaMKK3-Ab*	37.2
	Yannong 19	White-grained semi winter	3AS/4AL	*TaMFT-3Aa*/*TaMKK3-Ab*	71.6

## Data Availability

Not applicable.

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
