# Peer review of "Genetic Improvement of Wheat with Pre-Harvest Sprouting Resistance in China"

_genes, 2023, doi:10.3390/genes14040837_

Round 1

Reviewer 1 Report

Comments to the manuscript

Genetic improvement of wheat with pre-harvest sprouting resistance in China

Authors: Cheng Chang *, Haiping Zhang, Jie Lu, Hongqi Si, Chuanxi Ma *

Pre-harvest sprouting (PHS) is a very serious problem for wheat production in most wheat growth areas as rains before harvesting could severely deteriorate grain quality. The way to overcome this problem is breeding of cultivars with PHS resistance, which requires sources of such resistance as well as knowledge of its genetic basis for marker-assisted selection to increase the efficiency of breeding. In their review Chang et al. consider important aspects of PHS resistance: the most stable QTLs, cloned genes for PHS resistance and the situations of breeding of PHS-resistant cultivars in China. The Chinese experience and germplasm resources are of importance worldwide.

Some corrections are recommended for the manuscript.

In lane 56. “whole-genome association analysis based on natural populations...”

 natural populations are to be replaced by a panel of varieties

In Table 2 please add chromosome position (in Mb like in Table 1 or in cM) for the genes and references for favorable alleles.

In lane 171 Jining 13 is referred to as PHS-susceptible (GI: 32.3%) but in lanes 177 and 178  Aizap 781 (GI: 32.4%) and Dainong 64 (GI: 37.2%) are referred to as moderately resistant to PHS. How is it possible?

The text in lanes 205-210 repeats the information in lanes 81-90 and could be removed from 5.3

Lane 218 Please check the phrase ‘the later possibly has low effects’

Please add a supplementary table with the information on GI of the Chinese varieties mentioned in sections 4. and 5. Please fill this table with data on alleles of marker genes of PHS resistance if they are available in [7, 56-63].

References [5] and [8] are the same.

References [14] and [35] are the same.

Author Response

In lane 56. “whole-genome association analysis based on natural populations...”

 natural populations are to be replaced by a panel of varieties

Response: have been changed to a panel of varieties.

In Table 2 please add chromosome position (in Mb like in Table 1 or in cM) for the genes and references for favorable alleles.

Response: A good suggestion. We have added the local information of the genes and references.

In lane 171 Jining 13 is referred to as PHS-susceptible (GI: 32.3%) but in lanes 177 and 178  Aizap 781 (GI: 32.4%) and Dainong 64 (GI: 37.2%) are referred to as moderately resistant to PHS. How is it possible?

Response: Thanks  a lot for your suggestion, we have changed the mistake.

The text in lanes 205-210 repeats the information in lanes 81-90 and could be removed from 5.3

Response: We have removed the repeats.

Lane 218 Please check the phrase ‘the later possibly has low effects’

Response: We have revised the sentence.

Please add a supplementary table with the information on GI of the Chinese varieties mentioned in sections 4. and 5. Please fill this table with data on alleles of marker genes of PHS resistance if they are available in [7, 56-63].

Response: A good suggestion. We have added the information of varieties  in a supplementary table in revised manuscript.

References [5] and [8] are the same.

References [14] and [35] are the same.

Response: We have checked them and revised.

Reviewer 2 Report

A detailed review, but there are a number of inaccuracies and questions. 

Introduction: Line 25, the term monsoon is suitable only for part of the climatic zones, improve it.

Authors mentioning their research would require methodological explanations of how they were carried out or references to literary sources (English).

The review mentions many Chinese varieties. But no information was found on where to look for detailed information about them, as well as opportunities to fall on their authors.

The list of publications contains a series of layout inaccuracies: lines 306, 377, 380, 408, 410.

Author Response

Introduction: Line 25, the term monsoon is suitable only for part of the climatic zones, improve it.

Response: Thanks, we haved revised.

Authors mentioning their research would require methodological explanations of how they were carried out or references to literary sources (English).

Response: A good suggestion. We have added the references about the methods in the mentioning researches.

The review mentions many Chinese varieties. But no information was found on where to look for detailed information about them, as well as opportunities to fall on their authors.

Response: A very good suggestion, we have added a supplementary table of varieties' informations.

The list of publications contains a series of layout inaccuracies: lines 306, 377, 380, 408, 410.

Response: Thanks a lot, we have revised them.